# THE UPDATE-EQUIVALENCE FRAMEWORK FOR DECISION-TIME PLANNING

**Samuel Sokota**[†1]   **Gabriele Farina**[†2]

**David J. Wu**[†]   **Hengyuan Hu**[3]   **Kevin A. Wang**[†4]   **J. Zico Kolter**[1,5]   **Noam Brown**[†6]

[†]Work done at Meta AI  [1]Carnegie Mellon University  [2]Massachusetts Institute of Technology
[3]Stanford University  [4]Brown University  [5]Bosch AI  [6]OpenAI
ssokota@andrew.cmu.edu gfarina@mit.edu

## ABSTRACT

The process of revising (or constructing) a policy at execution time—known as *decision-time planning*—has been key to achieving superhuman performance in perfect-information games like chess and Go. A recent line of work has extended decision-time planning to *imperfect-information* games, leading to superhuman performance in poker. However, these methods involve solving subgames whose sizes grow quickly in the amount of non-public information, making them unhelpful when the amount of non-public information is large. Motivated by this issue, we introduce an alternative framework for decision-time planning that is not based on solving subgames, but rather on *update equivalence*. In this update-equivalence framework, decision-time planning algorithms replicate the updates of last-iterate algorithms, which need not rely on public information. This facilitates scalability to games with large amounts of non-public information. Using this framework, we derive a provably sound search algorithm for fully cooperative games based on mirror descent and a search algorithm for adversarial games based on magnetic mirror descent. We validate the performance of these algorithms in cooperative and adversarial domains, notably in Hanabi, the standard benchmark for search in fully cooperative imperfect-information games. Here, our mirror descent approach exceeds or matches the performance of public information-based search while using two orders of magnitude less search time. This is the first instance of a non-public-information-based algorithm outperforming public-information-based approaches in a domain they have historically dominated.

## 1 INTRODUCTION

Decision-time planning (DTP) is the process of revising (or even constructing from scratch) a policy immediately before using that policy to make a decision. In settings involving strategic decision making, the benefits of DTP can be quite large. For example, while decision-time planning approaches have achieved superhuman performance in chess (Campbell et al., 2002), Go (Silver et al., 2016), and poker (Moravčík et al., 2017; Brown and Sandholm, 2018; 2019), approaches without decision-time planning remain non-competitive.

Currently, the dominant paradigm for DTP is based on solving (or improving the policy as much as possible in) *subgames*. In perfect-information games, the subgame at a state is naturally defined as a game beginning from that state that proceeds according to the same rules as the original game. In contrast, the definition of subgame is nuanced in imperfect-information games, due to the presence of private information. While multiple definitions have been proposed, all those that facilitate sound guarantees (Nayyar et al., 2013; Dibangoye et al., 2013; Oliehoek, 2013; Burch et al., 2014; Moravcik et al., 2016; Brown and Sandholm, 2017; Moravčík et al., 2017; Brown et al., 2020; Sokota et al., 2023b) rely on variants of a distribution known as a *public belief state (PBS)*[1]—*i.e.*, the joint posterior over each player's decision point, given public information and the joint policy that has been played so far.

---

[1]Note that different literatures use different names to refer to this distribution and its variants.

Unfortunately, PBS-based planning has a fundamental limitation: it is ineffective in settings with large amounts of non-public information. This shortcoming arises because PBS-based DTP distributes its computational budget across all decision points supported by the PBS and because the number of such decision points increases with the amount of non-public information in the game. When the amount of non-public information is small, such as in Texas Hold'em, where there are only $\binom{52}{2} = 1326$ possible private hands, this is a non-issue because it is feasible to construct strong policies for decision points supported by the PBS. However, as the amount of non-public information grows, meaningfully improving for all such decision points becomes increasingly inviable. For example, in games where the only public knowledge is the amount of time that has passed, PBS-based subgame solving requires reasoning about every decision point consistent with the time step. In the worst case—games with no public information at all—PBS-based subgame solving requires solving the entire game, defeating the entire purpose of DTP.

In this work, motivated by the insufficiency of PBS-based conceptualizations, we investigate an alternative perspective on DTP that we call the *update-equivalence* framework for DTP. Rather than viewing DTP algorithms as solving subgames, the update-equivalence framework views them as implementing the updates of *last-iterate algorithms*—a term that we use to refer to algorithms whose last iterates possess desirable properties, such as monotonic improvements in expected return, decreases in exploitability, or convergence to an equilibrium point.[2] This framework offers a simple mechanism for generating and analyzing principled DTP algorithms, contrasting PBS-based algorithms and analyses, which are notorious, especially in adversarial games, for their reconditeness. Furthermore, importantly, because these last-iterate algorithms need not involve PBSs, the update-equivalence framework for DTP is not inherently limited by the amount of non-public information. Indeed, the most natural instances of the framework focus their entire computation budget on the decision point that is actually occupied by the planning agent, entirely ignoring counterfactual decision points supported by the PBS.

We invoke the update-equivalence framework to derive a DTP algorithm based on mirror descent (Nemirovsky and Yudin, 1983; Beck and Teboulle, 2003). We prove that this algorithm, which we call *mirror descent search (MDS)*, guarantees policy improvement in fully cooperative games. We test MDS in Hanabi (Bard et al., 2020), the standard benchmark for search in fully cooperative imperfect-information games. Remarkably, MDS exceeds or matches the performance of state-of-the-art PBS methods while using two orders of magnitude less search time. *This is the first instance of a non-public-information-based algorithm outperforming public-information-based approaches in a setting they've historically dominated.* Furthermore, we show that the performance gap between MDS and PBS methods widens as the amount of non-public information increases, reflecting MDS's superior scalability properties. In addition to MDS, we also introduce a search algorithm for adversarial settings based on magnetic mirror descent (MMD) (Sokota et al., 2023a) that we call *MMD search (MMDS)*. We show empirically that MMDS significantly improves performance in games where PBS methods are essentially inapplicable due to the large amount of non-public information.

## 2 BACKGROUND AND NOTATION

We use the formalism of finite-horizon partially observable stochastic games, or POSGs, for short (Hansen et al., 2004). POSGs are a class of games equivalent to perfect-recall timeable extensive-form games (Jakobsen et al., 2016; Kovařík et al., 2022). To describe finite-horizon POSGs, we use $s \in \mathbb{S}$ to notate Markov states, $a_i \in \mathbb{A}_i$ to notate player $i$'s actions, $o_i \in \mathbb{O}_i$ to notate player $i$'s observations, $h_i \in \mathbb{H}_i = \bigcup_t (\mathbb{O}_i \times \mathbb{A}_i)^t \times \mathbb{O}_i$ to notate $i$'s decision points, $a \in \mathbb{A} = \times_i \mathbb{A}_i$ to notate joint actions, $h \in \mathbb{H} = \times_i \mathbb{H}_i$ to notate histories, $\mathcal{R}_i \colon \mathbb{S} \times \mathbb{A} \to \mathbb{R}$ to notate player $i$'s reward function, $\mathcal{T} \colon \mathbb{S} \times \mathbb{A} \to \Delta(\mathbb{S})$ to notate the transition function, $\mathcal{O}_i \colon \mathbb{S} \times \mathbb{A} \to \mathbb{O}_i$ to notate player $i$'s observation function.

Each player $i$ interacts with the game via a policy $\pi_i$, which maps decision points to distributions over actions $\pi_i \colon \mathbb{H}_i \to \Delta(\mathbb{A}_i)$. Given a joint policy $\pi = (\pi_i)_i$, we use the notation $\mathcal{P}_\pi$ to notate

---

[2]For example, this definition of last-iterate algorithm would include policy iteration in Markov decision processes, since its last iterate converges to the optimal policy, but exclude algorithms for imperfect-information games possessing only average-iterate guarantees, such as counterfactual regret minimization (Zinkevich et al., 2007) and fictitious play (Brown, 1951).

probability functions associated with the joint policy $\pi$. For example, $\mathcal{P}_\pi(h \mid h_i)$ denotes the probability of history $h$ under joint policy $\pi$ given that player $i$ occupies decision point $h_i$.

The objective of player $i$ is to maximize its expected return $\mathcal{J}_i(\pi) := \mathbb{E}_\pi[G]$ where the return is defined as the sum of rewards $G := \sum_t \mathcal{R}_i(S^t, A^t)$ and where we use capital letters to denote random variables. (We maintain the convention of using capital letters for random variables throughout the paper.) If the return for each player is the same, we say the game is *common payoff*. If there are two players and the returns of these players are the opposite of one another, we say the game is *two player zero sum* (2p0s).

We notate the expected value for an agent at a history $h^t$ under joint policy $\pi$ as

$$v_i^\pi(h^t) = \mathbb{E}_\pi[G^{\geq t} \mid h^t] = \mathbb{E}_\pi\left[\sum_{t' \geq t} \mathcal{R}_i(S^{t'}, A^{t'}) \mid h^t\right].$$

The expected action value for an agent at a history $h^t$ taking action $a_i^t$ under joint policy $\pi$ is denoted

$$q_i^\pi(h^t, a_i^t) = \mathbb{E}_\pi[G^{\geq t} \mid h^t, a_i^t] = \mathbb{E}_\pi\left[\mathcal{R}_i(S^t, A^t) + v_i^\pi(H^{t+1}) \mid h^t, a_i^t\right].$$

The expected *value of a decision point $h_i^t$* is the weighted sum of history values, where each history is weighted by its probability, conditioned on the decision point and historical joint policy $v_i^\pi(h_i^t) = \mathbb{E}_\pi\left[v_i^\pi(H^t) \mid h_i^t\right]$. Similarly, the expected *action value for action $a_i^t$ at a decision point $h_i^t$* is defined as $q_i^\pi(h_i^t, a_i^t) = \mathbb{E}_\pi\left[q_i^\pi(H^t, a_i^t) \mid h_i^t\right]$.

## 3 THE FRAMEWORK OF UPDATE EQUIVALENCE

The framework of update equivalence rests on the idea that last-iterate algorithms and DTP algorithms may be viewed, implicitly or explicitly, as inducing updates. For last-iterate algorithms, this induced update refers to the mapping from one iterate (*i.e.*, policy) to the next. For DTP algorithms, this induced update refers to the mapping from the policy to be revised (called the blueprint policy) to the policy produced by the search. We define the idea that these updates may be equivalent below.

**Definition 3.1** (Update equivalence). For some fixed game, a decision-time planning algorithm and a last-iterate algorithm are *update equivalent* if, for any policy $\pi^t$ and any information state $h_i$, the distribution of actions played by the decision-time planning algorithm at $h_i$ with blueprint $\pi^t$ is equal to the distribution of actions the last-iterate algorithm would play at $h_i$ on iteration $t+1$ given that it had played $\pi^t$ on iteration $t$.

There are at least two benefits to considering update-equivalence relationships for decision-time planning. First, it begets an approach to analyzing DTP algorithms via their last-iterate algorithm counterparts. This avenue of analysis simply requires determining whether the next iterate is improved from the previous, circumventing a host of complications that make the analysis of PBS approaches nuanced. Second, and perhaps more importantly, it enables the generation of new principled DTP algorithms via last-iterate algorithms with desirable guarantees.

We illustrate these benefits in the two coming subsections. In Section 3.1, we focus on last-iterate algorithms that operate using action-value feedback, providing a general procedure to generate DTP analogues of these algorithms and discussing three DTP algorithms constructed via this procedure, as well as their soundness. In Section 3.2 we also discuss last-iterate algorithms that do not operate using action-value feedback, demonstrating how the framework of update equivalence can be used to generate or justify DTP algorithms with more complex structure.

### 3.1 ACTION-VALUE-BASED PLANNERS

In Algorithm 1, we give a procedure for constructing DTP algorithms that are update-equivalent to last-iterate algorithms operating on action-value feedback. Specifically, let the last-iterate algorithm be represented by a continuous function $\mathcal{U}: (\pi^t(h_i), q^{\pi^t}(h_i)) \mapsto \pi^{t+1}(h_i)$ mapping the current policy $\pi^t$ to the next policy $\pi^{t+1}$ by incorporating the action-value feedback $q^{\pi^t}$ for the current policy. Algorithm 1 repeatedly samples histories starting from a given decision point and the joint policy and acquires estimates of the sample returns for these histories via rollouts. When the computational

budget has been exhausted, Algorithm 1 applies the update function $\mathcal{U}$ to the current joint policy and estimated action values. This leads to an update-equivalent DTP algorithm, as we establish next.

**Proposition 3.2.** *Consider a last-iterate algorithm operating with action-value feedback whose update function $\mathcal{U}$ is continuous. Then, as the number of rollouts goes to infinity, the output of Algorithm 1, conditioned on $\mathcal{U}$ and given inputs $h_i^t, \pi$, converges in probability to $\mathcal{U}(\pi^t(h_i), q^{\pi^t}(h_i))(h_i)$.*

*Proof.* This proposition holds because $\hat{q}$ converges in probability to $q_i^\pi(h_i)$ by the law of large numbers and $\mathcal{U}$ is continuous. $\square$

**Monte Carlo Search** Algorithm 1 enables us to relate last-iterate algorithms with DTP counterparts. A notable existing instance of such a relationship had already been identified by Tesauro (1995) and Tesauro and Galperin (1996) between policy iteration (as the last-iterate algorithm) and Monte Carlo search (as the corresponding DTP algorithm). Indeed, by plugging policy iteration's local update function into Algorithm 1 we recover exactly Monte Carlo search as a special case. The local update induced by policy iteration can be written as

$$\mathcal{U} \colon (\bar{\pi}, \bar{q}) \mapsto \underset{\bar{\pi}' \in \Delta(\mathbb{A}_i)}{\arg\max} \langle \bar{\pi}', \bar{q} \rangle,$$

---

**Algorithm 1** Update Equivalent Search for Last-Iterate Algorithm with Action-Value Feedback and Update $\mathcal{U}$

---

**Input:** decision point $h_i^t$, joint policy $\pi$
Initialize $\bar{q}[a]$ as running mean tracker for $a \in \mathbb{A}_i$
**repeat**
    Sample history $H^t \sim \mathcal{P}_\pi(H^t \mid h_i^t)$
    **for** $a \in \mathbb{A}_i$ **do**
        Sample return $G^{\geq t} \sim \mathcal{P}_\pi(G^{\geq t} \mid H^t, a)$
        $\bar{q}[a]$.update_running_mean($G^{\geq t}$)
    **end for**
**until** search budget is exhausted
**return** $\mathcal{U}(\pi(h_i^t), \bar{q})$

---

where $\bar{\pi} \in \Delta(\mathbb{A}_i)$ and $\bar{q} \in \mathbb{R}^{|\mathbb{A}_i|}$. In other words, policy iteration's update simply plays a greedy policy with respect to its action values. By Proposition 3.2, when the computational budget given to the Monte Carlo search DTP algorithm grows, the policies produced converge to those of policy iteration. So, with enough budget, Monte Carlo search inherits all desirable properties enjoyed by policy iteration, including convergence to optimal policies in single-agent settings and convergence to Nash equilibrium in *perfect-information* 2p0s games (Littman, 1996).

**Mirror Descent Search** Algorithm 1 also enables us to give novel justification to DTP approaches in settings in which giving guarantees has been historically difficult. One example of such as a setting is common-payoff games, where a naïve application of Monte Carlo search can lead to bad performance as a result of the posterior over histories induced by the search policy diverging too far from the blueprint policy's posterior (Sokota et al., 2022). We show that this issue can be resolved using mirror descent (Beck and Teboulle, 2003; Nemirovsky and Yudin, 1983), which is a general approach to optimizing objectives that penalizes the distance between iterates. In the context of sequential decision making, a natural means of leveraging mirror descent is to instantiate simultaneous updaters $\mathcal{U}$ of the form

$$\mathcal{U} \colon (\bar{\pi}, \bar{q}) \mapsto \underset{\bar{\pi}' \in \Delta(\mathbb{A}_i)}{\arg\max} \langle \bar{\pi}', \bar{q} \rangle - \frac{1}{\eta}\mathrm{KL}(\bar{\pi}', \bar{\pi})$$

at each decision point, where $\bar{\pi}$ is a local policy, $\eta$ is a stepsize, $\bar{q}$ is a local action-value vector. In the case of discrete action spaces, this update reduces to the closed-form learning algorithm $\mathcal{U}(\bar{\pi}, \bar{q}) \propto \bar{\pi} e^{\eta \bar{q}}$, known in literature as *hedge* or *multiplicative weights update*.

In the appendix, we show that mirror descent satisfies the following improvement property.

**Theorem 3.3.** *Consider a common-payoff game. Let $\pi^t$ be a joint policy having positive probability on every action at every decision point. Then, if we run mirror descent at every decision point with action-value feedback, for any sufficiently small stepsize $\eta > 0$, $\mathcal{J}(\pi^{t+1}) \geq \mathcal{J}(\pi^t)$. Furthermore, this inequality is strict if $\pi^{t+1} \neq \pi^t$ (that is, if $\pi^t$ is not a local optimum).*

We call *mirror descent search (MDS)* the DTP algorithm obtained by applying Algorithm 1 with mirror descent as the last-iterate algorithm. By virtue of Proposition 3.2, as the computational budget increases, MDS inherits the improvement property of Theorem 3.3.

**Magnetic Mirror Descent Search** Another example of a setting that has been historically difficult for DTP is 2p0s games. In such settings, neither policy iteration- nor mirror descent-based approaches yield useful policies—instead, these approaches tend to cycle, diverge, or even exhibit

formally chaotic behavior already in simple games such as rock-paper-scissors (Cheung and Pil-iouras, 2019). It was recently shown that this issue can be empirically and reliably resolved using magnetic mirror descent (Sokota et al., 2023a), which is an extension of mirror descent with additional proximal regularization to a magnet that dampens these cycles. As with mirror descent, in the context of sequential decision making, a natural means of leveraging magnetic mirror descent is to instantiate simultaneous updaters $\mathcal{U}$ at each decision point:

$$\mathcal{U} \colon (\bar{\pi}, \bar{q}) \mapsto \underset{\bar{\pi}' \in \Delta(\mathbb{A}_i)}{\arg\max} \langle \bar{\pi}', \bar{q} \rangle - \frac{1}{\eta} \mathrm{KL}(\bar{\pi}', \bar{\pi}) - \alpha \mathrm{KL}(\bar{\pi}', \bar{\rho}),$$

where $\alpha$ is a regularization temperature and $\bar{\rho} \in \Delta(\mathbb{A}_i)$ is a local reference policy; in the case of discrete action spaces, this update reduces to the following closed form: $\mathcal{U}(\bar{\pi}, \bar{q}) \propto [\bar{\pi}e^{\eta\bar{q}}\bar{\rho}^{\eta\alpha}]^{\frac{1}{1+\alpha\eta}}$.

We call *magnetic mirror descent update equivalent search (MMDS)* the DTP algorithm obtained by applying Algorithm 1 with the particular choices of magnetic mirror descent as the last-iterate algorithm. Once again from Proposition 3.2, we establish that if the observation of MMD's reliable last-iterate convergence to equilibrium made by Sokota et al. (2023a) holds, then MMDS leads to expected improvement in 2p0s games as the computational budget grows.

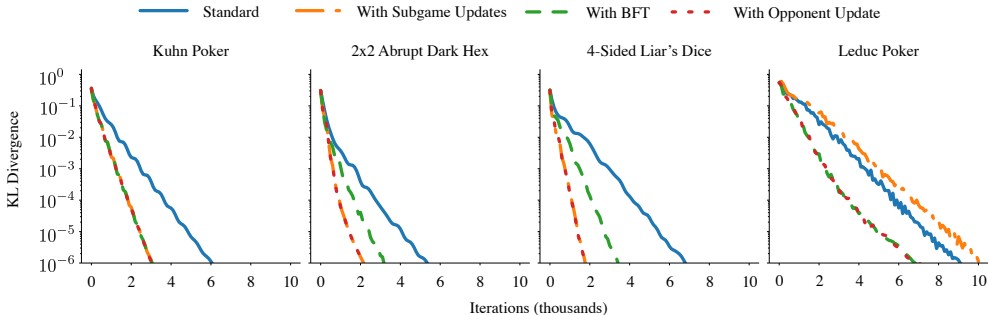

Figure 1: Divergence to AQRE as a function of iterations for last-iterate algorithm analogues of variants of MMD-based search algorithms. Each variant exhibits empirical convergence.

## 3.2 BEYOND ACTION-VALUE-BASED PLANNERS

Many DTP algorithms cannot be derived from Algorithm 1 that nevertheless may be useful to view from the perspective of update equivalence. One class of such approaches makes updates at future decision points during search. Approaches of this form, such as Monte Carlo tree search (MCTS) (Coulom, 2006; Kocsis and Szepesvári, 2006; Browne et al., 2012), played an important role in successes in perfect information games Silver et al. (2016; 2018); Schrittwieser et al. (2019); Antonoglou et al. (2022). Another class of approaches fine-tune the belief model (Sokota et al., 2022) to track the search policy.

We provide empirical evidence for the soundness of three such MMD-based approaches in the context of imperfect-information 2p0s games: 1) With *subgame updates*, A variant in which the planning agent also performs MMD updates at its own future decision points; 2) With *belief fine-tuning (BFT)*, a variant that implements an MMD update on top of a posterior fine-tuned to the search policy; 3) With *opponent updates*, a variant that implements an MMD update assuming a subgame joint policy in which the opponent has performed an MMD update at its next decision point. See Appendix B.1 for further algorithmic details.

In Figure 1, we measure the divergence of these last-iterate algorithm counterparts to agent quantal response equilibrium (AQRE) (McKelvey and Palfrey, 1998) as a function of the number of update iterations. We find that all three last-iterate algorithm analogues exhibit empirical convergence, suggesting that these DTP algorithms may be safe in 2p0s settings. More generally, these findings may hint that, so long as MMD is used as a local updater, one may have wide leeway in designing safe DTP algorithms for 2p0s games.

## 4 EXPERIMENTS

In this section, we add further evidence for the update-equivalence framework's utility by showing that the novel DTP algorithms derived from it also perform well in practice. We focus on two settings with imperfect information: i) two variants of Hanabi (Bard et al., 2020), a fully cooperative card game in which PBS-based DTP approaches are considered state-of-the-art; and ii) 3x3 Abrupt Dark Hex and Phantom Tic-Tac-Toe, 2p0s games with virtually no public information.

### 4.1 HANABI

We consider two versions of two-player Hanabi, a standard benchmark in the literature (Bard et al., 2020). Both versions are scored out of 25 points—scoring a 25 is considered winning. First, we trained instances of PPO (Schulman et al., 2017) in self play for each setting. We intentionally selected instances whose final performance roughly matched those of R2D2 (Kapturowski et al., 2019) instances that have been used to benchmark DTP algorithms. Next, we trained a belief model for each PPO policy using Seq2Seq with the same setup and hyperparameters as Hu et al. (2021); in this setup, the belief model takes in the decision point of one player and predicts the private information of the other player, conditioned on the PPO policy having been played thus far.

We adapted our implementation of MDS from that of single-agent SPARTA with a learned belief model (Hu et al., 2021). Our adaptation involves three important changes. First, MDS performs search for all agents, rather than only one agent, as was done in (Hu et al., 2021). Second, MDS plays the argmax[3] of a mirror descent update, rather than the argmax of the empirical Q-values from search, as SPARTA does. Third, MDS always plays its search policy. This contrasts both SPARTA (Lerer et al., 2020; Hu et al., 2021) and RLSearch (Fickinger et al., 2021), which, after doing search, perform a validation check to determine whether the search policy outperforms the blueprint policy; if the validation check fails, the blueprint policy is played instead of the search policy. This validation check can be expensive and usually fails (SPARTA and RLSearch only play their search policies on a handful of turns). Thus, MDS's ability to sidestep the validation check is a significant advantage.

**5-Card Hanabi** We first test MDS in the 5-card 8-hint variant of Hanabi, which is commonly played by humans. Specifically, we compare MDS using a PPO blueprint policy against single- and multi-agent SPARTA and RLSearch, which are considered state-of-the-art, with exact belief models and an R2D2 blueprint policy; we also compare against single-agent SPARTA using the same PPO blueprint policy and Seq2Seq belief model. For MDS, we use $10,000$ samples from the belief model for search; this takes about 2 seconds per decision, which is about two orders of magnitude faster than multi-agent SPARTA and multi-agent RL-Search.

We report our results in Table 1. To give context to the results, the strongest reported performance of model-free agents in literature exceeds 24.40, while that of the strongest reported search agents exceeds 24.60 (Hu et al., 2021). It becomes increasingly hard to improve the performance as the score increases; for high-scoring policies, increases as small as a few hundredths of points are considered substantial and can translate to multiple percentage points in winning percentage. To give specific numbers as an illustration, Lerer et al. (2020) report that improving the expected return from 24.53 to 24.61 can increase the winning percentage from 71.1% to 75.5%.

Despite being handicapped by 1) an approximate belief model, 2) a lack of validation, and 3) two orders of magnitude less search time, we find that MDS achieves superior or comparable performance to PBS methods. Given the historical dominance of PBS approaches in settings where they are applicable, the fact that a non-PBS-based method could achieve competitive performance is already notable; that MDS does so with such a severe handicap is particularly impressive.

**7-Card Hanabi** We next investigate MDS in the 7-card 4-hint variant of Hanabi. This variant was introduced by Hu et al. (2021) to illustrate the challenges of performing search as the amount of private information scales. Specifically, the number of possible histories is too large to enumerate. As a result, multi-agent SPARTA is inapplicable altogether, single-agent SPARTA and RLSearch require an approximate belief model, and multi-agent RLSearch requires both an approximate belief

---

[3]Note that this differs slightly from our description in the previous section in that we are playing the argmax, rather than sampling, from the updated policy.

Table 1: MDS (bold) compared to SPARTA and RLSearch in 5-card 8-hint Hanabi. Despite using approximate beliefs and roughly two orders of magnitude less search time, MDS matches the performance of prior methods using exact beliefs. Columns to the left of the divider used an R2D2 blueprint that scored 24.23±0.04; columns to the right used a PPO blueprint that scored 24.24±0.02.

| Search | SPARTA | RLSearch | SPARTA | RLSearch | SPARTA | **MDS** |
|---|---|---|---|---|---|---|
| Agents planned | Single | Single | Multi | Multi | Single | **Multi** |
| Belief model | Exact | Exact | Exact | Exact | Seq2Seq | **Seq2Seq** |
| Validation used | ✓ | ✓ | ✓ | ✓ | ✓ | **✗** |
| Time per move | 3s | 35s | 450s | 180s | 1s | **2s** |
| Expected return | 24.57 | 24.59 | 24.61 | 24.62 | 24.52 | **24.62** |
| Standard error | ± 0.03 | ± 0.02 | ± 0.02 | ± 0.03 | ± 0.02 | **± 0.02** |

Table 2: MDS (bold) compared to SPARTA and RLSearch in 7-card 4-hint Hanabi; MDS compares favorably to these approaches. Columns to the left of the divider used an R2D2 blueprint that scored $23.67 \pm 0.02$; columns to the right used a PPO blueprint that scored $23.66 \pm 0.03$.

| Search | RLSearch | RLSearch | RLSearch | SPARTA | **MDS** |
|---|---|---|---|---|---|
| Agents planned | Single | Single | Multi | Single | **Multi** |
| Belief model | Seq2Seq | BFT | BFT | Seq2Seq | **Seq2Seq** |
| Validation used | ✓ | ✓ | ✓ | ✓ | **✗** |
| Time per move | 35s | 100s | 310s | 1s | **2s** |
| Expected Return | 24.14 | 24.18 | 24.18 | 24.17 | **24.28** |
| Standard error | ± 0.04 | ± 0.03 | ± 0.03 | ± 0.03 | **± 0.02** |

model and belief fine-tuning (Sokota et al., 2022). Because of the significantly increased burden of running PBS methods in this setting, one would hope that a method that eschews PBSs might be able to outperform PBS methods. And, indeed, in Table 2, where we report expected return and standard error over 2000 games, we find that MDS does outperform multi-agent RLSearch with BFT, as well as a variety of single-agent search techniques. These results support our claim that the update-equivalence framework offers superior scaling properties in the amount of non-public information, compared to PBS methods.

To offer further intuition for the behavior of MDS, we show the performance in 7-card 4-hint Hanabi as a function of mirror descent's stepsize in Figure 2. As might be expected, we find that improvement is a unimodal function of stepsize: If the stepsize is too small, opportunities to improve the blueprint policy are neglected; on the other hand, if the stepsize is too large, the search policy's posterior diverges too far from that of the blueprint for the search to provide useful feedback.

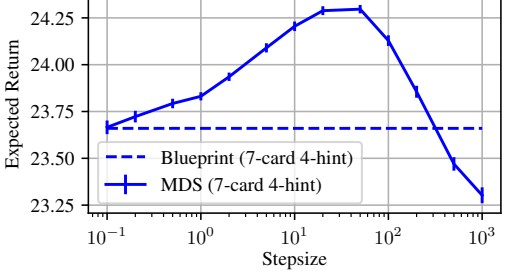

Figure 2: Performance of MDS as a function of stepsize in 7-card 4-hint Hanabi. Small step sizes provide less improvement over the blueprint; overly large step sizes cause the search policy to diverge too far from the blueprint, resulting in less improvement or even detriment.

## 4.2 3x3 ABRUPT DARK HEX AND PHANTOM TIC-TAC-TOE

Unlike common-payoff games, measuring performance in 2p0s games is difficult because the usual metric of interest—exploitability (*i.e.*, the expected return of a best responder)—cannot be cheaply estimated. Indeed, providing a reasonable lower bound requires training an approximate best response (Timbers et al., 2022), which may be costly. To facilitate this approximate best response training under a modest computation budget, we focus our experiments on a blueprint policy that selects actions uniformly at random at every decision point so that rollouts may be performed quickly; furthermore, rather than using a learned belief model, we track an approximate posterior using particle filtering (Doucet and Johansen, 2009) with 10 particles. If particles remain, our implementation of MMDS performs a rollout for each particle for each legal action until the end of the

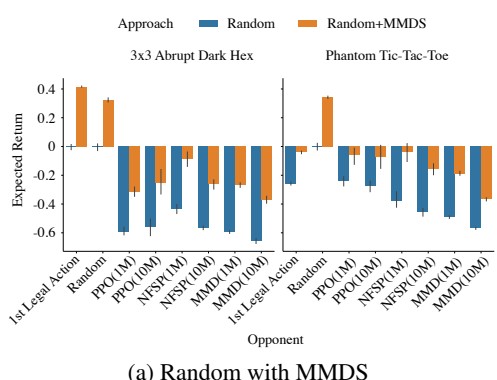 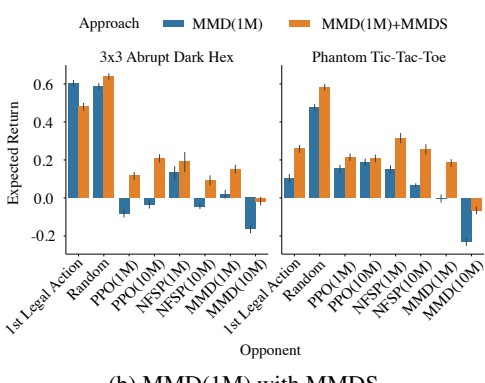

(a) Random with MMDS               (b) MMD(1M) with MMDS

Figure 3: Expected return of uniform random and uniform random blueprint + MMDS (left) and MMD(1M) and MMD(1M) blueprint + MMDS (right) versus various opponents in 3x3 Abrupt Dark Hex and Phantom Tic-Tac-Toe. MMDS tends to improve head-to-head expected return.

game and performs an MMD update on top of the empirical means of the returns. Otherwise, the agent executes the blueprint policy.

We empirically investigate MMDS in 3x3 Abrupt Dark Hex and Phantom Tic-Tac-Toe, two standard benchmarks available in OpenSpiel (Lanctot et al., 2019). We show approximate exploitability results in Table 3. We computed these results using OpenSpiel's (Lanctot et al., 2019) DQN (Mnih et al., 2015) best response code, trained for 10 million time steps. We show standard error over 5 DQN best response training seeds and 2000 final evaluation games for the fully trained model. We compare against a bot that plays the first legal action, the uniform random blueprint, Liang et al. (2018)'s implementation of independent PPO (Schulman et al., 2017), Lanctot et al. (2019)'s implementation of NFSP (Heinrich and Silver, 2016), and MMD (Sokota et al., 2023a). For the learning agents, we ran 5 seeds and include checkpoints trained for both 1 and 10 million time steps.

We find that MMDS reduces the approximate exploitability of the uniform random blueprint by more than a third, despite using a meager 10-particle approximate posterior. Furthermore, MMDS achieves lower approximate exploitability than all non-MMD-based approaches.

We also investigate the performance of MMDS in head-to-head matchups. We show results using the uniform random blueprint with 10 particles and also a MMD(1M) blueprint with 100 particles in Figure 3. The values shown are averages over 10,000 games with bootstrap estimates of 95% confidence intervals. We find that MMDS tends to improve the performance of the blueprint policies in both cases, despite the short length of the games.

Table 3: Approximate exploitability in 3x3 Abrupt Dark Hex and Phantom Tic-Tac-Toe, on a 0 to 100 scale. MMDS (bold) substantially reduces Random's exploitability.

| Agent | 3x3 Ab. DH | Phantom TTT |
|---|---|---|
| 1st Legal Action | $100 \pm 0$ | $100 \pm 0$ |
| Random | $74 \pm 1$ | $78 \pm 0$ |
| PPO(1M steps) | $85 \pm 6$ | $89 \pm 6$ |
| PPO(10M steps) | $100 \pm 0$ | $90 \pm 4$ |
| NFSP(1M steps) | $91 \pm 4$ | $95 \pm 1$ |
| NFSP(10M steps) | $59 \pm 1$ | $78 \pm 5$ |
| **Random+MMDS** | **$50 \pm 1$** | **$50 \pm 1$** |
| MMD(1M steps) | $34 \pm 2$ | $37 \pm 1$ |
| MMD(10M steps) | $20 \pm 1$ | $15 \pm 1$ |

## 5 RELATED WORK

**Other Approaches to Imperfect Information** Motivated by the deficiencies of PBS-based DTP described above, a small group of existing works has advocated for alternative approaches to DTP for common-payoff games (Tian et al., 2020) and 2p0s games (Zhang and Sandholm, 2021). Tian et al. (2020)'s approach relies on a result that shows that it is possible to decompose the change in expected return for two different joint policies across decision points. By leveraging this result, Tian et al. (2020) introduce a search procedure called joint policy search (JPS) that is guaranteed to not decrease the expected return. Zhang and Sandholm (2021)'s approach is based on the insight

that, in practice, it is effective to consider a subgame that excludes most of the decision points supported by the PBS. Specifically, Zhang and Sandholm (2021) advocate in favor of solving a maxmargin subgame (Moravcik et al., 2016) that includes the planning agent's true decision point, as well as any opponent decision points that are possible from the perspective of the planning agent. Despite proving the existence of games in which this approach, which they call 1-KLSS, increases the exploitability of the policy, Zhang and Sandholm (2021) find experimentally that, on small and medium-sized games, 1-KLSS reliably decreases exploitability. In concurrent work, Liu et al. (2023) derive an alternative variant of 1-KLSS that guarantees safety, making it a promising approach for scaling search to adversarial settings with large amounts of non-public information.

**Structurally Similar Approaches** While the motivation for the update-equivalence framework most closely resembles of the works of Tian et al. (2020) and Zhang and Sandholm (2021), natural instances of the update-equivalence framework are structurally more similar to search algorithms unrelated to resolving issues with PBS-based planning. As discussed previously, arguably the most fundamental instance of the framework of update equivalence is Monte Carlo search (MCS) (Tesauro, 1995; Tesauro and Galperin, 1996), which is update equivalent to policy iteration, as was articulated by Tesauro and Galperin (1996) themselves: "[MCS] basically implements a single step of policy iteration." More recently, Anthony et al. (2019); Anthony (2021); Hamrick et al. (2021); Lerer et al. (2020); Hu et al. (2021) investigate MCS in a variety of settings: Anthony et al. (2019) and Anthony (2021) find that it yields perhaps surprisingly good performance relative to MCTS in Hex; Hamrick et al. (2021) report comparable performance with MCTS across a variety of settings, with the exception of Go; Lerer et al. (2020) and Hu et al. (2021) show strong results for Hanabi (under the names single-agent SPARTA and learned belief search).

In his work, Anthony (2021) also investigates a search algorithm called policy gradient search that performs policy gradient updates at future decision points. He finds that a variant of this approach that involves regularization toward the blueprint policy tends to outperform not regularizing. This approach is similar to some of the variants of MMDS investigated in Section 3.2. The combination of policy gradient search and the framework of update equivalence may be a fruitful direction in the pursuit of algorithms that perform well in both perfect and imperfect information games.

Separately from Anthony (2021), Jacob et al. (2022) also investigate approaches to DTP involving regularization toward a blueprint. Jacob et al. (2022) show empirically that, by performing regularized search, it is possible to increase the expected return of an imitation learned policy without a loss of prediction accuracy (for the policy being imitated). The most immediately related experiments to this work are those concerning Hanabi, in which Jacob et al. (2022) investigated MDS (under the name piKL SPARTA) applied on top of imitation learned blueprint policies. Jacob et al. (2022) note that this approach reliably increases the performance of weak policies, but neither recognize that it possesses an improvement guarantee nor that it is simply performing a hedge update, as we do in this work. Jacob et al. (2022)'s experiments on Diplomacy (Paquette et al., 2019) are also related in that their approach is similar to a follow-the-regularized-leader analogue of MMDS. This approach played an important role in recent empirical successes for Diplomacy (Bakhtin et al., 2023; Meta Fundamental AI Research Diplomacy Team (FAIR) et al., 2022), suggesting that the framework of update equivalence is a natural approach to general-sum settings.

# 6 CONCLUSION AND FUTURE WORK

In this work, motivated by the deficiencies of search algorithms based on public belief state representation of imperfect-information games, we advocate for a new paradigm for decision-time planning, which we call the framework of update equivalence. We show how the framework of update equivalence can be used to generate and ground new decision-time planning algorithms for imperfect-information games. Furthermore, we show that these algorithms can achieve competitive (or superior) performance compared with state-of-the-art PBS-based methods in Hanabi and can reduce approximate exploitability in 3x3 Abrupt Dark Hex and Phantom Tic-Tac-Toe.

We believe the framework of update equivalence opens the door to many exciting possibilities for DTP and expert iteration (Anthony et al., 2017; Anthony, 2021) in settings with large amounts of imperfect information. Among these are the prospects of extending algorithms resembling AlphaZero (Silver et al., 2018), Stochastic MuZero (Antonoglou et al., 2022), and Diplodocus (Bakhtin et al., 2023) to imperfect-information games.

## 7 ACKNOWLEDGEMENTS

We thank Brian Hu Zhang for helpful discussions regarding knowledge-limited subgame solving (Zhang and Sandholm, 2021) and Eugene Vinitsky for helpful feedback regarding the presentation of the work.

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

# A  THEORY

In this section, we prove Theorem 3.3. To start, consider the following more general lemma.

**Lemma A.1** (Folklore). *Let $f \in \mathcal{C}^1(\mathcal{X})$, where $\mathcal{X} \subseteq \mathbb{R}^d$ is convex and compact, $x_0 \in \mathcal{X}$, and $x^+$ be the solution to the mirror descent step*

$$x^+ := \arg\min_{x \in \mathcal{X}} \{\eta\langle\nabla f(x_0), x\rangle + D_\varphi(x, x_0)\}.$$

*where $\varphi$ is differentiable and 1-strongly convex with respect to a norm $\|\cdot\|$. Then, for $\eta$ small enough, if $x^+ \neq x_0$,*

$$f(x^+) < f(x_0),$$

*Proof.* From the first-order necessary optimality conditions for the mirror descent step, we have

$$\langle\eta\nabla f(x_0) + \nabla\varphi(x^+) - \nabla\varphi(x_0), \hat{x} - x^+\rangle \geq 0 \quad \forall x \in \mathcal{X}.$$

Hence, letting $\hat{x} := x_0$, and using the strong convexity of $\varphi$ and the assumption that $x^+ \neq x_0$, we obtain

$$\langle\nabla f(x_0), x^+ - x_0\rangle \leq -\frac{1}{\eta}\|x^+ - x_0\|^2 < 0,$$

and with a further application of the Cauchy-Schwarz inequality,

$$\|x^+ - x_0\| \leq \eta\|\nabla f(x_0)\|.$$

By continuity of the gradient of $f$, there must exist $\epsilon > 0$ such that

$$\langle\nabla f(x), x^+ - x_0\rangle < 0 \quad \forall x \in B(x_0, \epsilon).$$

Furthermore, the mean value theorem guarantees that

$$f(x^+) - f(x_0) = \langle\nabla f(\xi), x^+ - x_0\rangle$$

for some $\xi$ on the line connecting $x_0$ to $x^+$. So, as long as $\eta\|\nabla f(x_0)\| < \epsilon$, we have

$$f(x^+) < f(x_0),$$

as we wanted to show. $\square$

**Theorem 3.3** Consider a common-payoff game. Let $\pi^t$ be a joint policy having positive probability on every action at each decision point. Then, if we run mirror descent at every decision point with action-value feedback, for any sufficiently small stepsize $\eta$,

$$\mathcal{J}(\pi^{t_1}) \geq \mathcal{J}(\pi^t).$$

Furthermore, this inequality is strict if $\pi^{t_1} \neq \pi^t$ (that is, if $\pi^t$ is not a local optimum).

*Proof.* The space of joint policies $\Pi$ is a Cartesian product of probability simplices, immediately implying that it is convex and compact. Furthermore, the expected return function is a polynomial function of any joint policy $\pi \in \Pi$. Hence, $\mathcal{J} \in \mathcal{C}^1(\Pi)$. The result then follows directly from Lemma A.1, as, by hypothesis, the players are performing mirror descent steps on $\mathcal{J}$. $\square$

# B  EXPERIMENTS

In this section, we provide further details about some of our empirical results and also show some additional results.

## B.1  BEYOND ACTION-VALUE-BASED PLANNERS

First, we describe in greater detail the DTP algorithms that we investigated in Section 3.2.

**With Subgame Updates**    With subgame updates differs from MMDS in the feedback it uses. In particular, rather than using the action values for the current policy, with subgame updates uses the action-values for the joint policy induced by performing MMD updates at its own future decision points (but leaving the opponent's policy fixed). Because the opponent is fixed, in tabular settings, we can compute the feedback for the last-iterate algorithm analogue of this approach using one backward induction pass for each player.

**With Belief Fine-Tuning**    With BFT differs from MMDS in the distribution it samples histories from. In particular, rather than sampling from the distribution induced by the current policy, it samples from the distribution induced by the search policy for each player.

**With Opponent Update**    With opponent update differs from MMDS in the feedback it uses. In particular, rather than using the action values for the current policy, with opponent update uses the action values for the joint policy induced by performing an MMD update for the opponent at the next time step. Note that, as a DTP algorithm, this approach would involve two belief model sampling steps: one to sample an opponent decision point for updating and one to sample an unbiased history for that opponent information state.

**Agent Quantal Response Equilibria Solving**    In Figure 4, we show the convergence results from the main body, along with the exploitabilities of the corresponding iterates. For these experiments, we used $\alpha = 0.1$ and $\eta = \alpha/10$, except for with opponent updates on Leduc, where we used $\eta = \alpha/20$, and with subgame updates on Leduc, where we used $\eta = \alpha/50$.

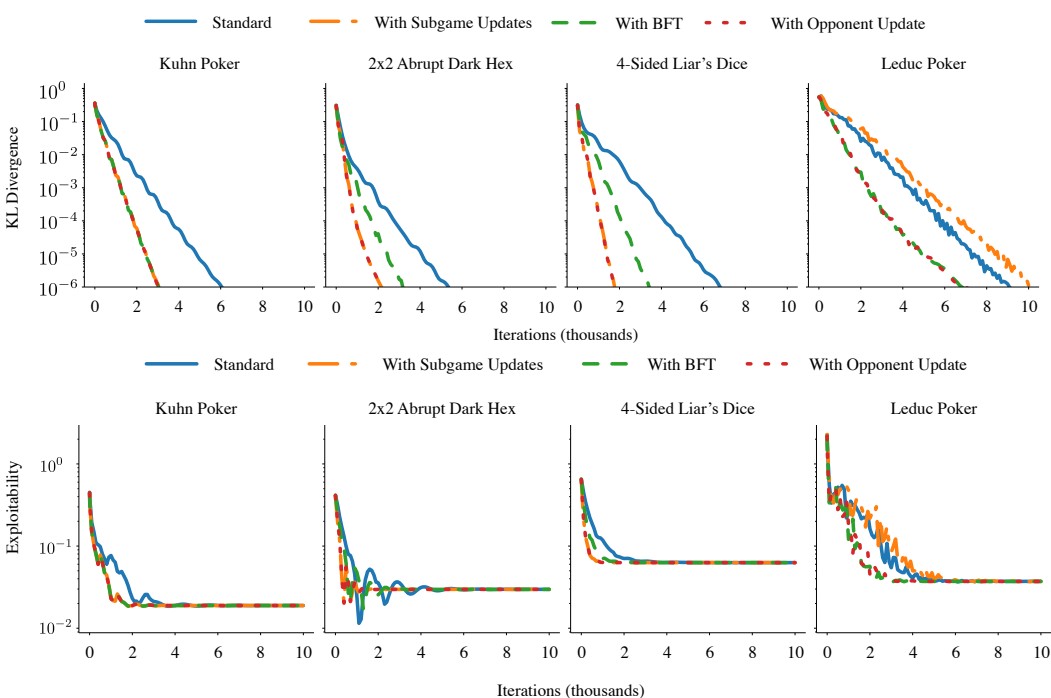

Figure 4: Solving for agent quantal response equilibria using last-iterate algorithm analogues of variants of MMDS.

**MiniMaxEnt Equilibrium Solving**    In Figure 5, we show convergence results for solving for MiniMaxEnt equilibria, which are the solutions of MiniMaxEnt objectives (Perolat et al., 2021). A MiniMaxEnt objective is an objective of the form

$$\mathcal{J}_i \colon \pi \mapsto \mathbb{E}\left[\sum_t \mathcal{R}_i(S^t, A^t) + \alpha\mathcal{H}(\pi_i(H_i^t)) - \alpha\mathcal{H}(\pi_{-i}(H_{-i}^t)) \mid \pi\right].$$

We used $\alpha = 0.1$ and $\eta = \alpha/10$, except for with opponent update on Leduc, where we used $\eta = \alpha/20$, and with subgame updates on Leduc, which used $\eta = \alpha/50$. We again observe empirical convergence.

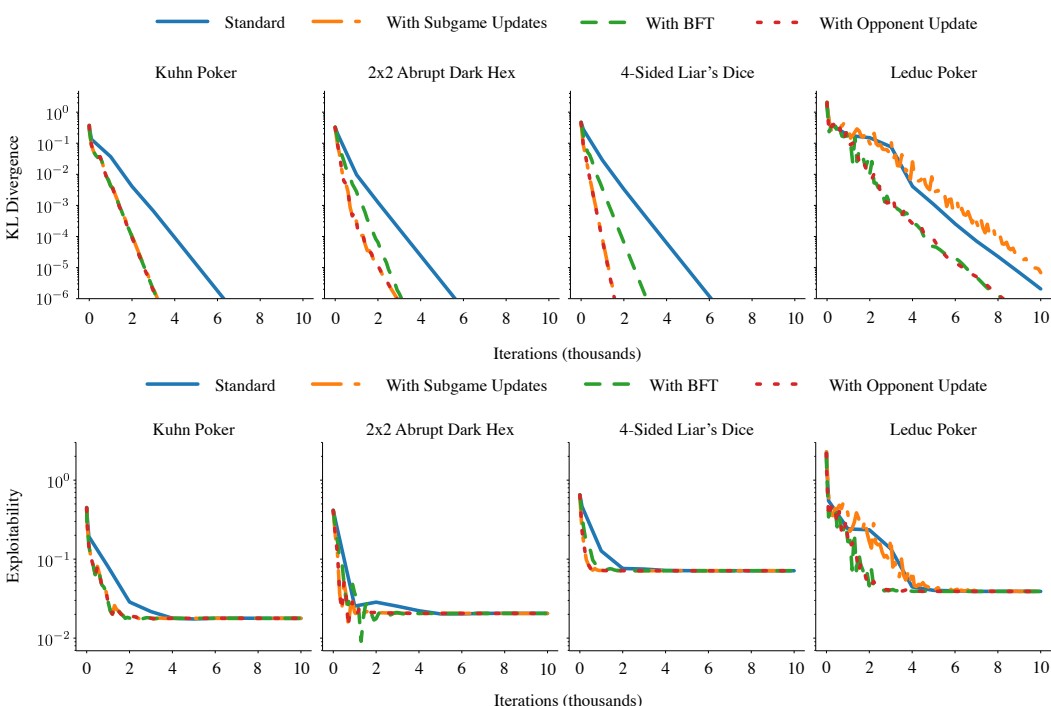

Figure 5: Solving for MiniMaxEnt equilibria using last-iterate algorithm analogues of variants of MMDS.

**Solving for Nash Equilibria** Next we show that these last-iterate algorithm analogues can be made to converge to Nash equilibria by annealing the amount of regularization used. We show that these results in Figure 6 compared against CFR (Zinkevich et al., 2007). The hyperparameters for these results are shown in Table 4.

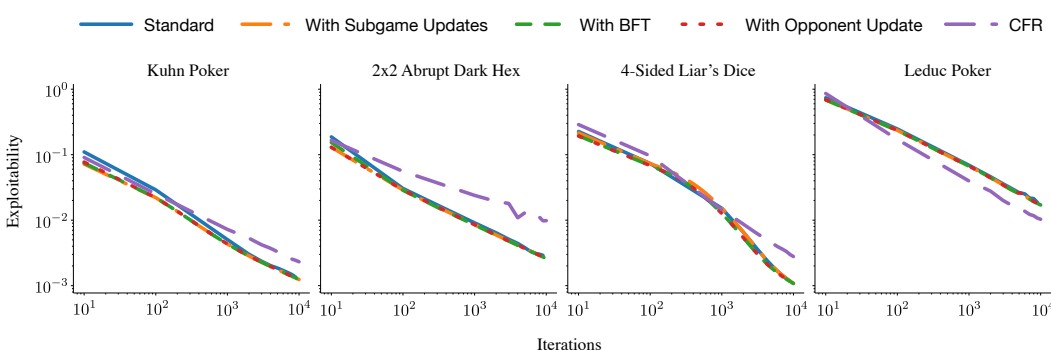

Figure 6: Exploitability of different MMDS analogues with annealed regularization.

| Method\Game | Kuhn Poker | 2x2 Abrupt Dark Hex | 4-Sided Liar's Dice | Leduc Poker |
|---|---|---|---|---|
| Standard | $\alpha_t = \eta_t = \frac{1}{\sqrt{t}}$ | $\alpha_t = \eta_t = \frac{1}{\sqrt{t}}$ | $\alpha_t = \frac{1}{\sqrt{t}}, \eta_t = \frac{2}{\sqrt{t}}$ | $\alpha_t = \frac{5}{\sqrt{t}}, \eta_t = \frac{1}{\sqrt{t}}$ |
| With Subgame Updates | $\alpha_t = \eta_t = \frac{1}{\sqrt{t}}$ | $\alpha_t = \eta_t = \frac{1}{\sqrt{t}}$ | $\alpha_t = \frac{1}{\sqrt{t}}, \eta_t = \frac{1}{2\sqrt{t}}$ | $\alpha_t = \frac{5}{\sqrt{t}}, \eta_t = \frac{1}{5\sqrt{t}}$ |
| With BFT | $\alpha_t = \eta_t = \frac{1}{\sqrt{t}}$ | $\alpha_t = \eta_t = \frac{1}{\sqrt{t}}$ | $\alpha_t = \frac{1}{\sqrt{t}}, \eta_t = \frac{2}{\sqrt{t}}$ | $\alpha_t = \frac{5}{\sqrt{t}}, \eta_t = \frac{1}{2\sqrt{t}}$ |
| With Opponent Update | $\alpha_t = \eta_t = \frac{1}{\sqrt{t}}$ | $\alpha_t = \eta_t = \frac{1}{\sqrt{t}}$ | $\alpha_t = \eta_t = \frac{1}{\sqrt{t}}$ | $\alpha_t = \frac{5}{\sqrt{t}}, \eta_t = \frac{1}{2\sqrt{t}}$ |

Table 4: Schedules for Figure 6.

**Solving for Nash Equilibria with MiniMaxEntRL objectives** We also show analogous results for MiniMaxEnt objectives in Figure 7. The hyperparameters for these experiments are shown in Table 5.

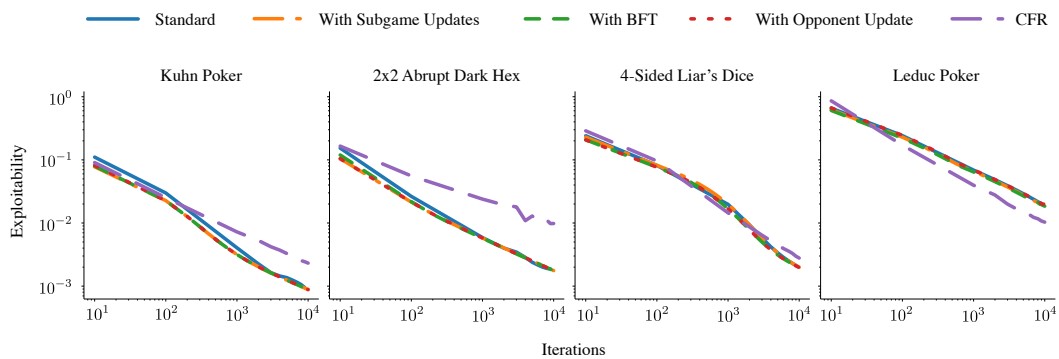

Figure 7: Exploitability of different MMDS analogues under MiniMaxEnt objectives with annealed regularization.

| Method\Game | Kuhn Poker | 2x2 Abrupt Dark Hex | 4-Sided Liar's Dice | Leduc Poker |
|---|---|---|---|---|
| One-Step Search | $\alpha_t = \eta_t = \frac{1}{\sqrt{t}}$ | $\alpha_t = \eta_t = \frac{1}{\sqrt{t}}$ | $\alpha_t = \frac{1}{\sqrt{t}}, \eta_t = \frac{2}{\sqrt{t}}$ | $\alpha_t = \frac{5}{\sqrt{t}}, \eta_t = \frac{1}{\sqrt{t}}$ |
| Multi-Step Search | $\alpha_t = \eta_t = \frac{1}{\sqrt{t}}$ | $\alpha_t = \eta_t = \frac{1}{\sqrt{t}}$ | $\alpha_t = \frac{1}{\sqrt{t}}, \eta_t = \frac{1}{2\sqrt{t}}$ | $\alpha_t = \frac{5}{\sqrt{t}}, \eta_t = \frac{1}{5\sqrt{t}}$ |
| BFT Search | $\alpha_t = \eta_t = \frac{1}{\sqrt{t}}$ | $\alpha_t = \eta_t = \frac{1}{\sqrt{t}}$ | $\alpha_t = \frac{1}{\sqrt{t}}, \eta_t = \frac{2}{\sqrt{t}}$ | $\alpha_t = \frac{5}{\sqrt{t}}, \eta_t = \frac{1}{\sqrt{t}}$ |
| Opponent Search | $\alpha_t = \eta_t = \frac{1}{\sqrt{t}}$ | $\alpha_t = \eta_t = \frac{1}{\sqrt{t}}$ | $\alpha_t = \eta_t = \frac{1}{\sqrt{t}}$ | $\alpha_t = \frac{5}{\sqrt{t}}, \eta_t = \frac{1}{2\sqrt{t}}$ |

Table 5: Schedules for Figure 7.

### B.2 Hanabi

For our Hanabi experiments, we used $\eta = 20$ for the MDS results in Tables 1 and 2. We performed search with 10,000 samples.

### B.3 3x3 Abrupt Dark Hex and Phantom-Tic-Tac-Toe

For our 3x3 Abrupt Dark Hex and Phantom-Tic-Tac-Toe experiments with a uniform blueprint, we used $\eta = 50$, $\alpha = 0.01$, and set $\rho$ to be uniform. For the MMD(1M) blueprint, used $\eta = 10$, $\alpha = 0.05$, and set $\rho$ to be uniform. For particle filtering, we sampled 10 particles for the uniform blueprint and 100 particles for the MMD(1M) blueprint from the start of the game independently at every decision point to reduce bias. For the baselines, we used the same setup as Sokota et al. (2023a).

