# OpenReview forum: "The Update-Equivalence Framework for Decision-Time Planning"
_ICLR.cc/2024/Conference — ICLR 2024 poster_

### Official Review · Reviewer_Vf98 · 2023-10-29

**Soundness:** 2 fair
**Presentation:** 1 poor
**Contribution:** 1 poor
**Rating:** 3
**Confidence:** 3

**Summary:**

The paper studies an equivalence between global policy learners and decision-time planning algorithms for imperfect information games. It argues that by using this equivalence, a new family of algorithms can be derived for solving imperfect information games that do not reply on public information. This makes the algorithms more efficient and more suitable for games where most information is non-public. The authors further tested the proposed approach in the Hanabi game, and demonstrated the advantage of the approach.

**Strengths:**

The paper seems to study an important problem relevant to interesting and high-impact applications of imperfect information games.

**Weaknesses:**

The paper is quite hard to follow. Through the current presentation, I don't find a clear message about the main approach and key idea of the work; See Questions. The notation and statements seem to lack clarity and rigor. The results presented look either confusing or trivial (e.g., Theorem 3.3).

**Questions:**

- What are the objectives of the algorithms (global policy learners and DTP algorithms)? Do they compute an equilibrium of the game, or do they just compute policies for the next iteration? What is the solution concept applied for the games studied?

- What does $g$ represent in $U^{global}: (\pi_t, g) \rightsquigarrow \pi_{t+1}$ when you introduce the global policy learners?

- $\pi_i$ is initially defined as the policy of player $i$, but when you introduce the global policy learners, $\pi_t$ seems to be associated with a time step $t$, so what does $\pi_t$ refer to here? Is there an iterative updating process where a new policy is used in each time step $t$?

- It is said that squiggly arrows are used to indicate randomness in the output. When you write $U_Q^{global} : \Delta(\mathbb{A}_i) \times \mathbb{R}^{|\mathbb{A}_i|} \rightsquigarrow \Delta(\mathbb{A}_i)$, the output is already a distribution over $\mathbb{A}_i$. Do you mean that there is a further randomization over the output distribution (i.e., a distribution over $\mathbb{A}_i$ is randomly chosen)?

- What is $\mathcal{P}_\pi$ in Algorithm 1? It seems undefined. What is $G$ samples in the algorithm?

---

> ### Author Response · Authors · 2023-11-20
> **Response to Reviewer Vf98**
>
> We thank the reviewer for their review.
>
> > What are the objectives of the algorithms (global policy learners and DTP algorithms)?
>
> The definitions of these classes of algorithms (global policy learners and DTP algorithms) specify the structure of the updates they induce, not their objectives. For global policy learners, the update is over the whole policy at once, whereas for DTP planning algorithms, it is only over a single decision-point.
>
> > Do they compute an equilibrium of the game
>
> Whether or not a global policy learner or DTP algorithm computes an equilibrium of the game depends on the algorithm and the game. For example, counterfactual regret minimization and policy gradient can both be considered as global policy learners. However, while counterfactual regret minimization computes equilibria in two-player zero-sum games, policy gradient does not (though it sometimes still performs well in practice).
>
> > do they just compute policies for the next iteration?
>
> Global policy learners compute policies for the next iteration. Decision-time planning algorithms compute policies for the current decision point.
>
> > What is the solution concept applied for the games studied?
>
> Solution concepts aren't necessarily the best way of thinking about the submission's framework in generality. The submission shows that decision-time planning algorithms can inherit the one-step guarantees of global algorithms. These guarantees can be related to solution concepts, but don't have to be. Of the algorithms studied in the experiments section, MD-UES possesses a guarantee on expected return improvement, while the results of Sokota et al. (2023) suggest that MMD-UES may possess guarantees related a decrease in exploitability. These guarantees are related to Nash equilibria in the sense that Nash equilibria are fixed points of the updates. However, at decision-time, we are only doing one-step improvements (and therefore not converging arbitrarily close to equilibria).
>
> > What does $g$ represent when you introduce the global policy learners?
>
> As is defined in Section 2, $g$ just represents the game itself (i.e., the states, actions, transition function, reward functions, etc.).
>
> > $\pi_i$ is initially defined as the policy of player $i$, but when you introduce the global policy learners, $\pi_t$ seems to be associated with a time step $t$, so what does $\pi_t$ refer to here?
>
> Thanks for pointing this out. $\pi_t$ refers to the joint policy at time $t$. We changed $\pi_t$ to $\pi^t$ in the text to clarify this issue.
>
> > Is there an iterative updating process where a new policy is used in each time step $t$?
>
> Yes, as is defined by the update function of a global policy learner update function in the text:
> $\mathcal{U}^{\text{global}} \colon (\pi_t, g) \mapsto \pi_{t+1}$.
>
> > Do you mean that there is a further randomization over the output distribution (i.e., a distribution over $\mathbb{A}_i$ is randomly chosen)?
>
> Yes.
>
> > What is $\mathcal{P}_{\pi}$ in Algorithm 1? It seems undefined.
>
> Thanks for pointing this out. $\mathcal{P}_{\pi}(h)$ is the probability of reaching history $h$ using joint policy $\pi$. We added this to the notation in Section 2 to clarify this issue.
>
> > What is $G$ samples in the algorithm?
>
> $G$ is the return, as is stated in the pseudocode text. We added this to the notation in Section 2 to clarify this issue.

---

### Official Review · Reviewer_W84D · 2023-10-31

**Soundness:** 3 good
**Presentation:** 2 fair
**Contribution:** 2 fair
**Rating:** 5
**Confidence:** 2

**Summary:**

This paper studies decision-time planning for general imperfect-information games, such as poker. Previous methods face limitations when dealing with games where the amount of non-public information is extensive, primarily due to the rapid growth in subgame sizes. To address this issue, this paper introduces a framework for decision-time planning, focusing on the idea of update equivalence rather than subgames. Experimental results demonstrate that these algorithms, when applied to Hanabi, 3x3 Abrupt Dark Hex, and Phantom Tic-Tac-Toe, either match or surpass state-of-the-art methods.

**Strengths:**

The main contribution of this work is proposing a new framework called "update equivalence" for decision-time planning in imperfect information games. The key idea is that instead of viewing DTP algorithms as solving subgames, they can be constructed to be equivalent to updates of global policy learners in the limit.

**Weaknesses:**

I find the proposed methods to be somewhat straightforward and intuitive, and I would encourage the authors to highlight the novelty and distinctiveness of their approach.

The theoretical results, such as Theorem 3.3, appear to be relatively basic and directly inferred from standard findings. Additionally, these results do not provide a comprehensive characterization of global convergence.


In terms of the empirical evaluation, I have reservations about the performance of the proposed methods, as they don't appear to offer a substantial improvement over existing algorithms. To strengthen the paper, it would be advantageous for the authors to broaden the scope of their evaluation, encompassing more complex scenarios.

**Questions:**

In terms of the definition of finite-horizon partially observable stochastic games, the observation function should be $\mathcal{O}_i: \mathbb{S} \rightarrow \mathbb{O}_i$ instead of $\mathcal{O}_i: \mathbb{S} \times \mathbb{A} \rightarrow \mathbb{O}_i$?

---

> ### Author Response · Authors · 2023-11-20
> **Response to Reviewer W84D**
>
> We thank the reviewer for their review.
>
> > I find the proposed methods to be somewhat straightforward and intuitive
>
> We are happy to hear that the reviewer finds the method straightforward and intuitive. However, that these methods are straightforward and intuitive are a strengh, not a weakness. It is important to appreciate once again that these methods outperform existing state-of-the-art methods for Hanabi, which are both much more complicated and much more expensive. So, pointing out that straightforward and intuitive methods are superior to involved constructions is an important observation for the literature (in our opinion).
>
> > I would encourage the authors to highlight the novelty and distinctiveness of their approach.
>
> The novelty is our recognition of the fact that update equivalence can justify principled decision-time planning algorithms in imperfect information games (indeed, no existing work recognized this fact!). As we state in the abstract: "Despite its conceptual simplicity, this approach had surprisingly been overlooked in the imperfect-information game literature." As a result of this novel recognition, we were able to achieve state-of-the-art performance on a competitive search benchmark with approaches far simpler and far less expensive than the previous state-of-the-art.
>
> Please let us know if we missed any opportunity to make this clear in the submission.
>
> > The theoretical results, such as Theorem 3.3, appear to be relatively basic and directly inferred from standard findings.
>
> The results ought not to be judged based on the difficulty of the proof but rather whether the result itself is important. Theorem 3.3 gives provable justification for update equivalence-based search in Dec-POMDPs. This allowed us to achieve a new state-of-the-art for Hanabi, which is the most competitive search benchmark for Dec-POMDPs.
>
> > Additionally, these results do not provide a comprehensive characterization of global convergence.
>
> This criticism could be applied to almost all literature on search in imperfect information games, so we do not feel that it is really a weakness.
>
> > In terms of the empirical evaluation, I have reservations about the performance of the proposed methods, as they don't appear to offer a substantial improvement over existing algorithms.
>
> As articulated in the submission, **the empirical results on Hanabi are very strong**. On 5-card Hanabi, we achieve comparable results to the existing SOTA with **two orders of magnitude** less search budget and an approximate belief (when we told an expert in Hanabi literature about these results, their initial reaction was that we must have a bug because the results seemed too good to be true). On 7-card Hanabi, we substantially outperform all existing methods, achieving a new state-of-the-art. We also show substantial improvement in imperfect-information settings in which public belief state-based methods are not even applicable.
>
> > To strengthen the paper, it would be advantageous for the authors to broaden the scope of their evaluation, encompassing more complex scenarios.
>
> Hanabi is both highly complex and the most well-benchmarked setting for search in large-scale Dec-POMDPs. There are multiple papers in literature that exclusively benchmark search results on Hanabi:
> - Improving Policies via Search in Cooperative Partially Observable Games (AAAI)
> - Scalable Online Planning via Reinforcement Learning Fine-Tuning (NeurIPS)
> - A Fine-Tuning Approach to Belief State Modeling (ICLR)
>
> Compared to these (published) works, we encompass a much wider variety of settings by also including adversarial settings.
>
> ---
>
> Please let us know whether these answers have resolved the reviewer's concerns. If so, we would appreciate the reviewer considering the possibility of revising their assessment.

---

> > ### Comment · Reviewer_W84D · 2023-11-22
> >
> > Thank you for your response. After reviewing the author's response and taking into account the comments from the other reviewers, I decide to maintain my original score.

---

### Official Review · Reviewer_ipeG · 2023-11-01

**Soundness:** 2 fair
**Presentation:** 1 poor
**Contribution:** 2 fair
**Rating:** 3
**Confidence:** 3

**Summary:**

This paper studies the decision-time planning problem with imperfect information by relating it to global policy learners. It introduce an update equivalence framework, based on which two algorithms are proposed to turn global policy learners to decision time planners. Experiments on games validate the promising performances of the proposed algorithms.

**Strengths:**

+ Gaming with imperfect information is a challenging problem, and this work provides an approach that is different from the conventional Public Belief State (PBS)-based planning.
+ The experiment results seem promising.

**Weaknesses:**

The presentation of this work is poor, which makes it very hard to understand the core idea of this work and assess its soundness and contribution. Specific examples include:
- Several notations are never defined, such as $S^t$, $A^t$, $H^{t+1}$, $h_i^t$, $h_i$, etc.
- The definitions of global policy learner and decision-time planning are not clear.
- Proposition 3.2 shows the connection between $\mathcal{U}^{global}$ and $\mathcal{U}^{global}_Q$. However, it is unclear how could Algorithm 1 converts a global policy learner to a decision-time planner, as claimed in the paper.
-  What does it mean by policy iteration local updating function? How to plug it into Algorithm 1? Why it can turn the algorithm to Monte Carlo search?
- What is the definition of $\sigma$ and how to integrate the constraint on $\sigma$?

**Questions:**

- Does Proposition 3.2 rely on the definition of global policy learner operating with action-value feedback?
- How to guarantee the desired connection between $\mathcal{U}^{global}$ and $\mathcal{U}^{global}_Q$ exist, and if so, how to obtain the corresponding function?

---

> ### Author Response · Authors · 2023-11-20
> **Response to Reviewer ipeG**
>
> We thank the reviewer for their review.
>
> > Several notations are never defined, such as $S^t, A^t, H^{t+1}, h_i^t, h_i$, etc.
>
> The listed notations listed by the reviewer are defined in Section 2 of the submission:
> - "$s \in \mathbb{S}$ to notate Markov states"
> - "$a \in \mathbb{A} = \times_i \mathbb{A}_i$ to notate joint actions"
> - "$h \in \mathbb{H} = \times_i \mathbb{H}_i$ to notate histories"
> - "$h_i \in \mathbb{H}_i = \cup_t (\mathbb{O}_i \times \mathbb{A}_i)^t \times \mathbb{O}_i$ to notate i’s decision points"
> - "we use capital letters to denote random variables"
>
> > The definitions of global policy learner and decision-time planning are not clear.
>
> Could the reviewer specify exactly what is unclear?
>
> > Proposition 3.2 shows the connection between $\mathcal{U}^{\text{global}}$ and $\mathcal{U}_Q^{\text{global}}$. However, it is unclear how could Algorithm 1 converts a global policy learner to a decision-time planner, as claimed in the paper.
>
> Algorithm 1 is a decision-time planning algorithm (by construction) and takes the local update function of a global policy learner $\mathcal{U}_Q^{\text{global}}$ as input (by construction).
>
> > What does it mean by policy iteration local updating function?
>
> We don't use the term "policy iteration local updating function" in the submission. We do use the term "policy iteration's local update function". This is defined in the text:
> "The local update induced by policy iteration can be written as $U_Q^{\text{global}} \colon (\bar{\pi}, \bar{q}) \mapsto \sigma$ where $\Sigma_{a \in \text{arg max } \bar{q}} \sigma(a) = 1$, where $\bar{\pi}$ and $\bar{q}$ refer to generic elements of $\Delta(\mathbb{A}_i)$ and $\mathbb{R}^{|\mathbb{A}_i|}$, respectively."
>
> > How to plug it into Algorithm 1?
>
> $U^{\text{global}}_Q$ is applied on the final line of the pseudocode, as is written in the text.
>
> > Why it can turn the algorithm to Monte Carlo search?
>
> As is discussed in the text, Algorithm 1, given policy iteration's $U^{\text{global}}_Q$, coincides with Monte Carlo search. Both estimate action values using rollouts and select an action with a maximal action value.
>
> > how to integrate the constraint on $\sigma$?
>
> There is no constraint on $\sigma$ that requires integration. It is just a distribution over the maximally valued actions.
>
> > Does Proposition 3.2 rely on the definition of global policy learner operating with action-value feedback?
>
> Yes, because Proposition 3.2 makes is a statement about global policy learners operating with action-value feedback it is inherently tied to the definition of global policy learners operating with action-value feedback.
>
> > How to guarantee the desired connection between $\mathcal{U}^{\text{global}}$ and $\mathcal{U}^{\text{global}}_Q$ exist, and if so, how to obtain the corresponding function?
>
> There is no such guarantee. The reviewer may misunderstand the purpose of this connection. It's not to show this connection exists for every $\mathcal{U}^{\text{global}}$ (it doesn't); it's to exploit this connection for specific $\mathcal{U}^{\text{global}}$ for which the connection does exist. For example, the desired connection exists for mirror descent and magnetic mirror descent, which allowed us to derive the search algorithms stuied in the paper.

---

> > ### Comment · Reviewer_ipeG · 2023-11-22
> >
> > I thank the authors for the response. However, I find the current version of this paper is still very hard to follow. It may require significant efforts to improve the clarity of the presentation throughout the paper, so that the intellectual merit of this work can become clearer. I decide to keep my current rating.

---

### Official Review · Reviewer_kQSj · 2023-11-01

**Soundness:** 3 good
**Presentation:** 3 good
**Contribution:** 3 good
**Rating:** 6
**Confidence:** 2

**Summary:**

The paper studies decision-time planning (DTP) which is crucial in achieving superhuman performance in games, especially in imperfect-information games. The authors introduce the concept of update equivalence, a new framework for DTP that replicates the updates of global policy learners. This approach allows for sound and effective decision-time planning in games with extensive non-public information. The authors propose two DTP algorithms, mirror descent update equivalent search (MD-UES) and magnetic mirror descent update equivalent search (MMD-UES), and evaluate their performance in various games.

**Strengths:**

1. The introduction of the notion of update equivalence is novel, which is not constrained by the amount of non-public information, making it applicable to a wider range of imperfect-information games.
2. The proposed algorithm  presents competitive or superior results compared to state-of-the-art subgame-based methods while requiring significantly less search time.

**Weaknesses:**

1. The requirement that algorithm needs to exhaust all the search budget seems inefficient and the performance highly depends on this computational costs.
2. I would prefer a separate "related works" section to make the presentation more clear.

**Questions:**

How does the proposed framework of update equivalence fundamentally differ from the traditional subgame-based approach in DTP? Can authors put the differences or advantages into several points so that the ideas are clear?

---

> ### Author Response · Authors · 2023-11-20
> **Response to Reviewer kQSj**
>
> We thank the reviewer for their review.
>
> > The requirement that algorithm needs to exhaust all the search budget seems inefficient and the performance highly depends on this computational costs.
>
> There may be a misunderstanding about the usage of terminology here. In search literature, it is common to say "until search budget is exhausted" to mean "do search for as long as is convenient". There is no requirement to fully exhaust any particular budget. Indeed, the class of algorithms we empirically investigate fall under the class "anytime algorithms" which is a term that means that they can be stopped at any time.
>
> > I would prefer a separate "related works" section to make the presentation more clear.
>
> To clarify, there is already a separate related works section (Section 5) with about a page of content.
>
> > How does the proposed framework of update equivalence fundamentally differ from the traditional subgame-based approach in DTP?
>
> Conceptually they can be seen as doing two different things. Subgame-based approaches define an auxiliary game and try to solve that game as best as possible. Update equivalence approaches try to approximate a pre-defined update step.
>
> An advantage of subgame-based approaches is that one can sometimes achieve a larger improvement by solving the entire subgame than would be possible by performing a single update step. An advantage of update equivalence-based approaches is that can you sidestep the need to construct a subgame, which can be expensive or intractable in imperfect information settings. For example, even in 7-card Hanabi, it becomes very difficult to employ public belief state-based methods because the public belief state is too large to compute exactly.

---

> > ### Comment · Reviewer_kQSj · 2023-11-23
> >
> > Thanks for the responses. After reevaluating the paper and reading other reviewers' comments, I agree that the presentation need to improve. As the update equivalence notion seems new to me, I decide to lower my score, but keep it above the threshold.

---

### Author Response · Authors · 2023-11-20
**General Message to Reviewers**

We thank all of the reviewers for their reviews. We continue to believe in the strength of the submission, in large part due to the empirical results in Hanabi, where we show that novel, simple, computationally inexpensive, scalable algorithms perform comparably or superior to state-of-the-art approaches, which are more complicated, more computationally expensive, and lack scalability. We believe that we have addressed all of the questions posed by the reviewers. We would appreciate if the reviewers could update their evaluations and scores, clearly articulating any aspect they still don't like about the paper so that we can continue improving it for the future.

---

### Meta-Review · Area_Chair_svdk · 2023-12-06

**Metareview:**

(a) Summarize the scientific claims and findings of the paper:

The paper introduces a novel framework for decision-time planning (DTP) in imperfect information games, emphasizing the concept of "update equivalence." This framework is designed to replicate updates of global policy learners, offering an alternative to traditional subgame-based approaches. The authors propose two algorithms under this framework: Mirror Descent Update Equivalent Search (MD-UES) and Magnetic Mirror Descent Update Equivalent Search (MMD-UES). These algorithms are evaluated in various games, including Hanabi, showcasing their effectiveness compared to state-of-the-art methods. The authors claim that the proposed algorithms are simpler, computationally less expensive, and more scalable while delivering comparable or superior performance.

(b) Strengths of the paper:

(+) Novelty of Update Equivalence (Reviewer kQSj, Reviewer W84D): The introduction of the concept of update equivalence is a novel contribution.  Despite its conceptual simplicity, this approach had surprisingly been overlooked in the imperfect-information game literature. The relaxation of the size of non-public information makes it applicable to a broader range of imperfect information games.

(+) Competitive Performance with Less Search Time (Reviewer kQSj, Reviewer W84D): The proposed algorithms show superior performance compared to existing state-of-the-art methods while requiring significantly less search time.

(+) Applicability to Imperfect Information Games (Reviewer ipeG, Reviewer W84D): The paper looks at the more challenging setting of imperfect information games, different from conventional Public Belief State (PBS)-based planning.

(+) Clarity and Intuitiveness of Methods (Reviewer W84D): The method is straightforward and intuitive. The fact that mirror descent leads to update equivalence seems elegant and useful.

(c) Weaknesses of the paper:

(-) Presentation and Clarity (Reviewer ipeG, Reviewer Vf98): Reviewers pointed out the presentation of the paper was poor. Notations were reportedly undefined or unclear, and the core idea was hard to follow. The explanation of how the algorithms convert a global policy learner to a decision-time planner was specifically cited as unclear.

(-) Lack of Comprehensive Theoretical Characterization (Reviewer W84D): The theoretical results, such as Theorem 3.3, are seen as relatively basic and lacking a comprehensive characterization of global convergence.

(-) Limited Scope of Empirical Evaluation (Reviewer W84D): While the Hanabi results are impressive, demonstrating the algorithm on a broader set of imperfect information games would strengthen the paper.

**Justification For Why Not Higher Score:**

The paper has the makings for a spotlight, if the exposition had been more accessible. For instance, while I am very familiar with planning, I am not as well versed on imperfect information games, and I had a little bit of trouble following the arguments. The rewrites improved the paper significantly, but I think there's room for improvement. An intuitive running example would help a lot.

**Justification For Why Not Lower Score:**

Given the significance of the results on Hanabi, one of the most challenging Dec-POMDP problems, this paper certainly should not be rejected. I think this could be a foundational paper for future works in imperfect information games.

---

### Decision · Program_Chairs · 2024-01-16

Accept (poster)